# Selecting the Most Effective DBS Contact in Essential Tremor Patients Based on Individual Tractography

**DOI:** 10.3390/brainsci10121015

**Published:** 2020-12-20

**Authors:** Jan Niklas Petry-Schmelzer, Till A. Dembek, Julia K. Steffen, Hannah Jergas, Haidar S. Dafsari, Gereon R. Fink, Veerle Visser-Vandewalle, Michael T. Barbe

**Affiliations:** 1Department of Neurology, Faculty of Medicine and University Hospital Cologne, University of Cologne, 50923 Köln, Germany; jan.petry-schmelzer@uk-koeln.de (J.N.P.-S.); julia.steffen@uk-koeln.de (J.K.S.); hannah.jergas@uk-koeln.de (H.J.); haidar.dafsari@uk-koeln.de (H.S.D.); gereon.fink@uk-koeln.de (G.R.F.); michael.barbe@uk-koeln.de (M.T.B.); 2Cognitive Neuroscience, Institute of Neuroscience and Medicine (INM-3), Research Center Jülich, 52428 Jülich, Germany; 3Department of Stereotactic and Functional Neurosurgery, Institute of Medicine and University Hospital Cologne, University of Cologne, 50923 Köln, Germany; veerle.visser-vandewalle@uk-koeln.de

**Keywords:** essential tremor, dentatorubrothalamic tract, volume of tissue activated, deep brain stimulation, tractography, automated programming

## Abstract

Postoperative choice of the most effective deep brain stimulation (DBS) contact in patients with essential tremor (ET) so far relies on lengthy clinical testing. Previous studies showed that the postoperative effectiveness of DBS contacts depends on the distance to the dentatorubrothalamic tract (DRTT). Here, we investigated whether the most effective DBS contact could be determined from calculating stimulation overlap with the individual DRTT. Seven ET patients with bilateral thalamic deep brain stimulation were included retrospectively. Tremor control was assessed for each contact during test stimulation with 2mA. Individual DRTTs were identified from diffusion tensor imaging and contacts were ranked by their stimulation overlap with the respective DRTT in relation to their clinical effectiveness. A linear mixed-effects model was calculated to determine the influence of the DRTT overlap on tremor control. In all investigated DBS leads, the contact with the best clinical effect was the contact with the highest or second-highest DRTT-overlap. At the group level, the DRTT-overlap explained 26.7% of the variance in the clinical outcomes (*p* < 0.001). Our data suggest that the overlap with the DRTT based on individual tractography may serve as a marker to determine the most effective DBS contact in ET patients and reduce burdensome clinical testing in the future.

## 1. Introduction

Essential tremor (ET) is the most common adult movement disorder, causes significant disability, interferes with activities of daily living, and reduces quality of life. For medication-refractory cases, deep brain stimulation (DBS) of the thalamic ventral intermediate nucleus (VIM) and the posterior subthalamic area (PSA) is an established, effective, and safe treatment [1,2]. However, postoperative choice of the most effective contact relies on time-consuming and exhausting clinical testing, especially with new generations of “directional leads” consisting of up to eight contacts.

DBS most likely modulates pathologic activity within the tremor network via cerebello-thalamo-cortical connections, i.e., the dentatorubrothalamic tract (DRTT) [3]. Additionally, it has been shown that direct targeting of the DRTT leads to successful tremor control and that the effectiveness of a contact depends on its distance to the DRTT [3,4,5,6]. We hypothesized that the most effective contact can be determined in silico by calculating the overlap of the stimulation with the respective DRTT.

## 2. Materials and Methods

### 2.1. Study Design

This retrospective study included ET patients who had received bilateral stereotactic DBS lead implantation into the PSA/VIM between January 2019 and January 2020. Only patients for whom preoperative structural magnetic resonance imaging (MRI), preoperative diffusion tensor imaging (DTI), and postoperative computed tomography (CT) scans were available were included in this study. Patient selection and implantation procedures have been described in detail [2]. All patients were implanted with directional leads (Cartesia^TM^, Boston Scientific, Middlesex County, MA, USA). Three months after implantation, patients underwent routine clinical testing to determine the most effective contact for postoperative tremor control. Following the Declaration of Helsinki, the study protocol was approved by the local ethics committee (Vote: 20-1511). Due to the retrospective character of the study, no informed consent was needed.

### 2.2. Clinical Outcome and Lead Reconstruction

As per clinical routine at our center, postural tremor, intention tremor, and rest tremor of the upper limb contralateral to active stimulation were assessed during a contact-wise stimulation with a fixed amplitude of 2 mA, a frequency of 130 Hz, and a pulse width of 60 µs as well as during the “OFF stimulation” baseline. Contralateral stimulation was switched off during the testing. For the directional levels, each directional contact was examined separately. Postural, intention, and rest tremor were each scored from 0 (“no tremor”) to 4 (“most severe tremor”) based on the “Tremor Rating Scale” [7], after a stimulation wash-in period of 1 to 2 min. The percentage change in the sum of these scores compared to the OFF stimulation baseline was calculated and used to rank the overall effectivity of DBS contacts. DBS leads and their respective rotations were identified from postoperative CT scans, and lead locations were transformed into the preoperative MRI using the Lead-DBS toolbox (https://www.lead-dbs.org/) [8,9,10]. Respective volumes of tissue activated (VTAs) were calculated in individual patient space using FASTFIELD with an electrical field threshold of 0.2 V/mm and an isotropic conductivity of 0.1 S/m [11,12].

### 2.3. Probabilistic Tracking of the DRTT

MRI data were acquired on a 3-Tesla Philips Ingenia® Scanner (Philips, Amsterdam, The Netherlands) (T1-sequence—TR: 9.8 ms; TE: 4.9 ms; acquisition time: 6.13 min; voxel-size: 0.49 × 0.49 × 1.00mm³). For diffusion imaging, a single-shot 2D, spin-echo, echo-planar imaging pulse sequence was applied (TR: 8213 ms; TE: 103 ms; 40 gradient directions; b-value: 1000 s/mm²; acquisition time: 9:53 min; voxel-size; 2.0 × 2.0 × 2.0 mm³). For probabilistic fiber tracking, we used the FMRIB software library (FMRIB, Oxford, UK). We employed probabilistic fiber tracking as it might be better at detecting the DRTT than the deterministic algorithms embedded in commercially available stereotactic planning software [13]. Diffusion data were corrected for susceptibility-induced distortions using the topup tool and corrected for head motion and Eddy current distortion using the eddy tool. Brain extraction of the b0 image was performed using BET, and distributions of diffusion vectors were estimated for each voxel with BEDPOSTX. The number of fibers per voxel was set to two. Probabilistic fiber tracking was performed separately for each DRTT with PROBTRACKX2 using a modified Euler integration. Streamlines were forced to cross waypoints in the selected order. For all other parameters, the respective default settings were used. The choice of regions of interest has been described previously [5]. In brief, the contralateral dentate nucleus was chosen as a seed region, while the contralateral superior cerebellar peduncle, the ipsilateral red nucleus, and the ipsilateral precentral gyrus served as waypoints. These regions of interest were previously defined in MNI space and were transformed to individual diffusion space using SPM (http://www.fil.ion.ucl.ac.uk/spm/software/spm12/) [5]. The resulting track frequency maps were visually examined for anatomical accuracy and transformed into a track probability map, according to Schlaier et al. [13]. Finally, the resulting fiber tracts were co-registered to the preoperative T1.

### 2.4. Statistical Analysis

The overlap of each 2 mA VTA with the respective DRTT was calculated as the sum of the resulting track probability map values in each voxel covered by the respective VTA, multiplied with the VTA’s voxel size in mm³. These DRTT overlap values were ranked hemisphere-wise to determine the contact with the largest DRTT overlap. We then investigated how often the electrodes with the highest and second-highest overlap also had the best tremor suppression during clinical testing. Additionally, a linear mixed-effects model was employed to determine the predictive value of the overlap with the individual DRTTs regarding tremor control at the group level. We included “DRTT overlap” as the main effect and “lead” as a random-effect to take multiple testings per lead into account. [14].

### 2.5. Data Availability

MATLAB scripts are available at the open science framework via the doi: 10.17605/OSF.IO/KXAMJ. Anonymized imaging and clinical data are available upon request to the corresponding author and are not publicly available due to privacy concerns.

## 3. Results

A total of seven ET patients (three female, age 68.8 y ± 14.8, disease duration: 26.1 y ± 13.7) and 14 directional DBS leads were included in this retrospective analysis. Two patients were taking medications in addition to DBS at 3-MFU (one patient propranolol 160 mg/day, one patient primidone 250 mg/day). Results of contact ranking by clinical effectiveness and overlap with the individual DRTT are shown in Figure 1. In 64.3% of the cases (9 of 14 hemispheres), the contact with the highest overlap with the individual DRTT showed the best clinical outcome or was among those with the best outcome if more than one contact showed equal tremor improvement. In 5 of 5 of the remaining hemispheres, the contact with the second-highest overlap with the individual DRTT showed the best clinical outcome. When only investigating directional contacts, in 57.1% of the cases, the directional contact with the highest DRTT overlap also had the best tremor improvement. At the group level, the linear mixed-effects model explained 68.4% of the variance of clinical outcome, while the overlap with the individual DRTT alone (main-effect) explained 26.7% of the variance (R²_model_ = 0.684, R²_main-effect_ = 0.267, *p* < 0.001, see Figure 2). A positive relationship between DRTT overlap and tremor improvement was observed in all hemispheres.

## 4. Discussion

This study demonstrates that overlap with the DRTT might serve as a marker for in silico determination of the most effective contact for tremor suppression in ET patients. The overlap with the DRTT determined one of the most effective contacts in 64.3% of cases. When also considering the contact with the second-highest overlap, this increased to 100% of cases. In other words, if one had only interrogated the two contacts with the highest DRTT overlap, a contact with an optimal outcome would have been determined in each hemispheres. With new generations of DBS leads, there is the option of steering the current towards more effective contacts, away from contacts causing side effects [15]. Only considering directional contacts, the chance of activating the most effective directional contact without clinical testing increases from 16.7% (1 out of 6) to 57.1% when using the in silico approach presented here.

Although landmark-based targeting was used when implanting our patients [2], the most ventral contact was the most effective contact with the highest overlap to the DRTT in the majority of cases. This finding is in line with previous studies indicating that (i) effective contacts are located inside or close to the DRTT [3,5], and (ii) that in our targeting approach, contacts in the PSA are closer to the DRTT [5]. While there was a positive relationship between DRTT overlap and tremor suppression in all hemispheres in our mixed-effects model, and overlap predicted 26.7% of the variance in tremor outcome, there were still marked individual differences between hemispheres, as indicated by the 68% of variance explained when also considering hemisphere as a random effect. As shown in Figure 1, leads with less tremor suppression showed a displacement from the DRTT.

Several attempts to predict postoperative tremor suppression have focused on connectivity analysis or probabilistic stimulation mapping [16,17,18]. However, only Åström et al. [19] focused on predicting the DBS contact to be chosen postoperatively. Based on probabilistic stimulation maps, the resulting software tool showed that the predicted contact with rank 1 matched the clinically used contact in 60% of cases (rank 1–2 matched 83% of the cases). In contrast, the present study was based on the individual DRTT as a neuroanatomical correlate of tremor suppression and included leads with eight contacts instead of four contacts.

This study’s major limitation is that we only investigated tremor suppression and did not consider stimulation-induced side effects. In several cases, more than one contact showed equally good tremor suppression. In such cases, side-effect thresholds would be crucial for determining the contact used for clinical stimulation. Therefore, more research regarding the neuroanatomical origins of different stimulation-induced side effects [20] is needed. Prospective studies also taking stimulation-induced side effects such as muscle contractions, paresthesia, ataxia, and stimulation-induced dysarthria into account should be conducted to validate and extend this retrospective analysis. Another important limitation is the use of individual diffusion imaging, which might not be available in many centers. On the other hand, there is mounting evidence for the involvement of the DRTT in treating tremor, and we thus assume that diffusion imaging and direct targeting of the DRTT will be more widely used in the near future.

## 5. Conclusions

As a conclusion, our study clearly demonstrates how in silico imaging analysis could guide clinical DBS programming in ET and help reduce patient burden by shortening tedious monopolar review investigations.

## Figures and Tables

**Figure 1 brainsci-10-01015-f001:**
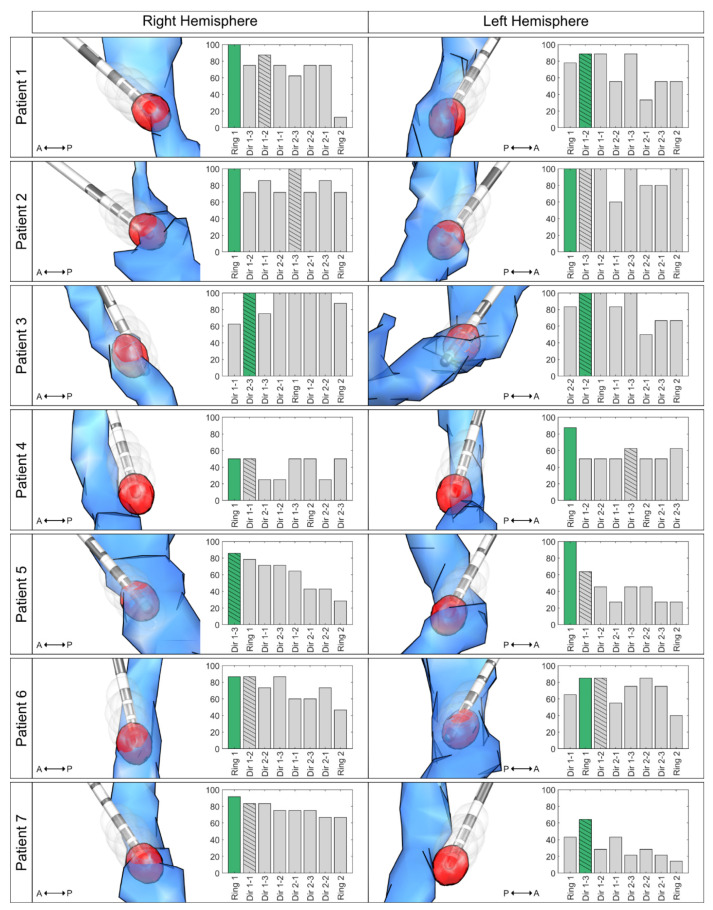
Contact Ranking. Bar plots illustrate the ranking of the overlap, with the individual dentatorubrothalamic tract (DRTT) per contact on the x axis (highest overlap to lowest overlap) and the improvement in tremor control in % on the y axis. The most effective contact is marked in green (with the highest overlap in cases where more than one contact had the best improvement), and the most effective directional contact bar is hatched. The respective left column illustrates the relation between the generated volumes of tissue activated (VTAs, gray) and the DRTT (blue) and the respective lead in the medial view. The VTA with the highest DRTT overlap is highlighted in red. For illustration, only the 10% highest values of the track probability map are shown. Abbreviations: A = anterior; DRTT = dentatorubrothalamic tract; P = posterior; VTA = volume of tissue activated; Dir 1 = ventral directional level; Dir 2 = dorsal directional level.

**Figure 2 brainsci-10-01015-f002:**
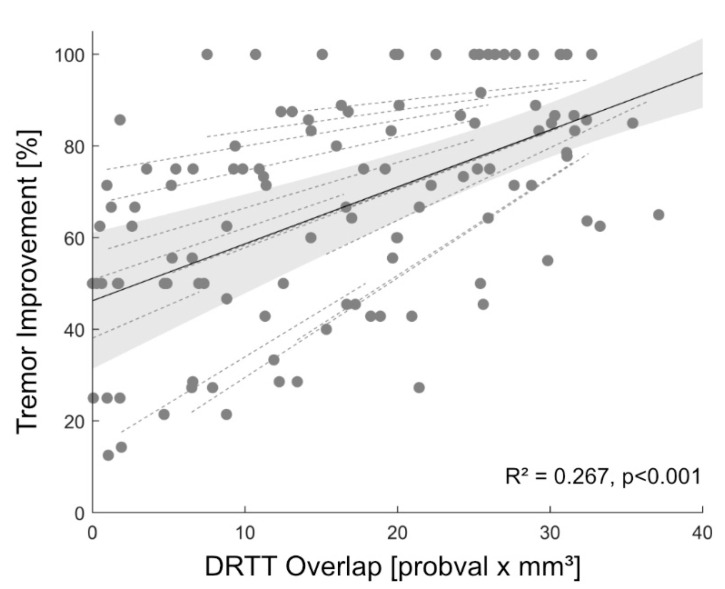
Prediction of Tremor Improvement. Linear mixed-effects model (black) and 95% confidence interval (gray) between tremor improvement and overlap with the individual dentatorubrothalamic tract (DRTT). Random effects for each individual hemisphere are also shown (dashed, gray). Abbreviations: DRTT = dentatorubrothalamic tract.

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
