# Peer review of "Selecting the Most Effective DBS Contact in Essential Tremor Patients Based on Individual Tractography"

_brainsci, 2020, doi:10.3390/brainsci10121015_

Round 1

Reviewer 1 Report

This brief paper sought to test the hypothesis that the most effective contact for VIM DBS to treat ET can be determined computationally by the overlap of the stimulation parameters only with individual DRTTs. The utility of this method for selecting effective directional contacts is notable, given the complexity of programming directional leads. Overall, only minor suggestions are made to improve clarity of the work presented and additional areas for discussion.

  • Clarify how tremor control/improvement was calculated. If correct, this change would suffice:
    • Tremor improvement was calculated using the sum of postural, intention, and rest tremor percentual change scores from the “OFF stimulation” baseline, which was used for the contact ranking and linear mixed-effects model analyses.
  • When “for analysis” is used, it is confusing. Specify which analysis/analyses.
  • There are grammatical and punctuation errors that should be corrected throughout.
  • Figure 1 is very informative – lots of data are visualized/depicted.
  • It may be worth commenting on the left hemisphere results for patient 7 – the most effective contact had lowest response (compared to the other hemispheres/patients) which may be attributed to displacement from the DRTT.
  • While these results do suggest that this method may shorten the monopolar review for directional leads, it relies on acquiring DTI and probabilistic fiber tracking which are not readily available/accessible or easy to do for the standard practitioner. Consider commenting on future clinical utility of this method, given this consideration.

Author Response

We thank the Reviewer for their assessment of our manuscript and their constructive criticism. We addressed all their comments as follows:

Clarify how tremor control/improvement was calculated. If correct, this change would suffice:

Tremor improvement was calculated using the sum of postural, intention, and rest tremor percentual change scores from the “OFF stimulation” baseline, which was used for the contact ranking and linear mixed-effects model analyses.

Reply: We have rephrased the section accordingly:

"(Methods) Postural, intention, and rest tremor were each scored from 0 (“no tremor”) to 4 (“most severe tremor”). The percentual change in the sum of these scores compared to “OFF stimulation” baseline was calculated and used to rank the overall effectivity of DBS contacts."

When “for analysis” is used, it is confusing. Specify which analysis/analyses.

Reply: We have rephrased our manuscript throughout to avoid this wording.

There are grammatical and punctuation errors that should be corrected throughout.

Reply: We hopefully routed out most of these errors during another round of careful proofreading.

Figure 1 is very informative – lots of data are visualized/depicted.
It may be worth commenting on the left hemisphere results for patient 7 – the most effective contact had lowest response (compared to the other hemispheres/patients) which may be attributed to displacement from the DRTT.

Reply: We added this important point to our manuscript:

"(Figure legend) Of note, DBS leads with worse clinical outcome (e.g. patient 4 right, patient 7 left) were further away from the DRTT."

"(Discussion) As shown in Figure 1, leads in which less tremor suppression was observed showed a displacement from the DRTT."

While these results do suggest that this method may shorten the monopolar review for directional leads, it relies on acquiring DTI and probabilistic fiber tracking which are not readily available/accessible or easy to do for the standard practitioner. Consider commenting on future clinical utility of this method, given this consideration.

Reply: Again the reviewer raises an important issue which we have addressed in our discussion:

"(Discussion) Another important limitation is the use of individual diffusion imaging, which might not be available in many centers. On the other hand there is mounting evidence for the involvement of the DRTT in treating tremor and we thus assume that diffusion imaging and direct targeting of the DRTT will be more widely used in the near future."

Reviewer 2 Report

In this paper, the authors present a retrospective study of essential tremor patients with bilateral PSA/VIM DBS and use diffusion tensor imaging data to determine the proximity of each DBS contact to the dentatorubrothalamic tract. The level of tremor control achieved by each contact is compared to the degree of volume of tissue activated which overlaps with the dentatorubrothalamic tract. In the majority of cases, the most effective contact exhibited the greatest degree of overlap with the DRTT.

The main question addressed in this paper is whether proximity to the dentatorubrothalamic tract could be used to predict the clinically most effective contact in essential tremor patients with PSA/VIM DBS. This is particularly relevant, as DBS systems are becoming increasingly more complex, putting greater time and knowledge burden on the clinicians that program them. Therefore, employing automized methods or tools could greatly help clinicians to program these devices in shorter amounts of time. If effective, this could also serve to reduce the time it takes for patients to experience the greatest degree of clinical benefit. Additionally, automized tools may further enable the use of DBS in underserved areas which lack specialists with extensive programming experience. Clinicians with experience in DBS will find this paper of particular interest.

A recent paper from the same group of authors demonstrated that efficiency of PSA and VIM DBS in essential tremor depends on the distance to the DRTT. This paper adds a logical next step to demonstrate that the clinically most effective contacts are usually those that have the greatest degree of overlap with the DRTT and propose that this may be used to predict the most effective contact. While similar studies have been performed in Parkinson’s Disease and overlap with the sensorimotor STN, to my knowledge, this is the first study to evaluate degree of tremor efficacy based on degree of overlap with DRTT.

The paper is well written and easy to comprehend. Figure 1 is particularly useful, as it effectively visualizes the data on an individual patient level in a clear manner.

The paper provides sufficient evidence to suggest that proximity of DBS contacts could be used as possible predictor of the most effective DBS contacts and does support their proposed hypothesis. As they mention in their discussion, the influence of side effects must be further explored and would be an important next step in their work.

Author Response

We thank the Reviewer for their generous assessment of our work.